# High-dimensional phenotyping reveals novel macrophage-like and hybrid subsets within murine splenic conventional dendritic cells

Chunqing Yang[1]*, Qingjie Xue[2], Yu Feng[1], Wenjun Ding[1], Ying Lu[3], Qinqin Wang[1]*

**1** Institute of Mental Health, Jining Medical University, Shandong, China, **2** Institute of Basic Medicine, Jining Medical University, Shandong, China, **3** Department of Endocrinology, East Hospital of the First People's Hospital of Jining, Shandong, China

* qqwang@mail.jnmc.edu.cn (QQW); ycqangel@126.com (CQY)

## Abstract

Conventional dendritic cells (cDCs) are pivotal antigen-presenting cells (APCs) with critical roles in immune regulation, yet their subset classification remains ambiguous due to phenotypic overlap with macrophages and monocytes, particularly in the spleen. This study employed multi-parametric flow cytometry and clodronate liposome (CL) depletion to systematically re-evaluate splenic CD11c$^{high}$MHCII$^{high}$ cDCs in C57BL/6 mice. We identified three novel subsets: (1) a tissue-resident T-cell zone macrophage (TZM)-like population (F4/80$^{inter-low}$CX3CR1$^+$MERTK$^+$) constituting 0.59% of cDC2s with >10-fold CL-depletion resistance ($p < 0.0001$); (2) a resident F4/80$^{high}$ APC subset (CCR2$^-$Ly6C$^-$) accounting for 2.7% of cDC2s with CL-sensitivity; (3) unconventional CD4$^+$CD8α$^+$ hybrids present in 2.57% of cDC2 and some cDC1s. These findings demonstrate unprecedented cDC plasticity driven by microenvironmental signals, revising conventional classification frameworks and proposing new targets for DC-based immunotherapies in autoimmunity and cancer. Our phenotypic mapping provides a foundational framework for future functional investigations into these novel subsets.

## 1 Introduction

Conventional dendritic cells (cDCs) are highly heterogeneous antigen-presenting cells (APCs) [1–4], and their subset classification still faces challenges because of their phenotypic overlap with that of macrophages and monocytes [2,3,5]. Recent single-cell transcriptome studies have suggested transitional differentiation phenomena among cDC subsets [6], yet the phenotypic plasticity of splenic cDC populations and their immune regulatory significance require further elucidation.

The spleen, as the largest secondary lymphoid organ in the body, harbors functionally specialized cDC subsets [2,7,8]. In steady-state mouse spleens, cDCs are primarily categorized into three types: (1) cDC1s, which are regulated by the

**Data availability statement:** All relevant data are within the manuscript and its Supporting Information files.

**Funding:** This work was supported by the Natural Science Foundation of Shandong Province (ZR2024MH129), the National Nature Science Foundation of China (82301629) and Project of the Innovative Training Program for Students at Jining Medical University (cx2020118). There was no additional external funding received for this study.

**Competing interests:** The authors have declared that no competing interests exist.

Irf8→Nfil3→Id2→Batf3 transcriptional cascade [3,9–11], characteristically express XCR1, CLEC9A, and CD8α [2,5,12], and specialize in cross-presenting antigens to CD8+ T cells, which are crucial for antiviral and antitumor immunity [2,5,13]; and (2) cDC2s, which demonstrate marked heterogeneity and are governed by Zeb2/Relb transcription factors [3,10]. On the basis of differences in ESAM and T-bet/ROR-γt expression, cDC2s can be further divided into ESAM+ cDC2A (T-bet+, promoting T helper 2 (Th2) immune responses) and ESAM<low/-> cDC2B (ROR-γt+, driving T helper 17 (Th17) cell differentiation) [2–5,7]; (3) cDC3s, an independent lineage distinct from cDC1/cDC2s, are derived from Ly6C+ monocyte–DC progenitors (MDPs) and differentiate under the regulation of CSF1 and the transcription factors Klf4/Irf8, which specifically express molecules such as Lyz2 and CLEC12A and exhibit a functional advantage in Th17 polarization [2,14].

Notably, within the CD11c<high>MHCII<high> cDC population, multiple cell subsets with overlapping phenotypes exist, such as a) migratory macrophage/cDC2 hybrid F4/80<high> APCs, which coexpress markers of tissue-resident macrophages (F4/80, CD64, MERTK) and cDC2-specific molecules (CD11c, MHCII, CD11b, and CD24) while possessing the antigen cross-presentation ability of cDC1s [15]; b) T-cell zone macrophages (TZMs) in the lymph nodes, which express CX3CR1 and MERTK but lack F4/80 and CD169, clear apoptotic cells through a CX3CR1-dependent pathway, and are resistant to clodronate liposome (CL) depletion [16]; and c) nonclassical cDC subsets, including CD103-CD11b- Sirpα+CX3CR1+CD8α+ cDC2-like cells [17] and mature cDC1-expressing cDC2-related markers [18]. These findings reveal the high phenotypic plasticity of cDC populations: in addition to classical cDC1s, cDC2s, and cDC3s, transitional cells with myeloid cell characteristics also exist. Given the architectural similarity between the splenic white pulp and the lymph node T-cell zone (TCZ) [7], we hypothesized that novel macrophage subsets analogous to TZMs may reside within the splenic TCZ. Furthermore, the conventional gating strategy (CD11c<high>MHCII<high>) likely fails to capture phenotypically ambiguous populations [2,16]. This knowledge gap underscores the necessity of integrating single-cell transcriptomics with spatial resolution to establish a refined cDC taxonomy [6,15]. However, the existence and phenotypic features of analogous transitional populations in splenic cDCs remain incompletely defined.

In this study, we conducted a systematic dissection of the cellular heterogeneity within the splenic cDC compartment. Our work reveals previously unrecognized macrophage-like cell subsets, a diphenotypic cDC population displaying cDC1/cDC2 transitional features, and nonclassical cDC subsets with distinct surface marker profiles. These findings provide critical insights into DC subset plasticity, offering a foundation for refining DC classification frameworks and revealing potential targets for DC-based immunotherapies.

## 2 Materials and methods

### 2.1 Animals

SPF-grade female C57BL/6 mice (7–8 weeks old) were purchased from Jinan Pengyue Experimental Animal Company. Mice were euthanized using gradual-fill carbon dioxide ($CO_2$) inhalation followed by secondary physical confirmation (cervical

dislocation). $CO_2$ was administered at a flow rate displacing 30%−70% of the chamber volume per minute to minimize distress, in accordance with AVMA guidelines. Prior to $CO_2$ exposure, mice were not anesthetized, as this method is deemed humane and compliant with institutional ethical standards. The experiments were approved by the Laboratory Animal Center Ethics Committee of Jining Medical University (JNMC-2024-DW-184).

## 2.2 Reagents and antibodies

FBS, EDTA, ACK lysing buffer, type VI collagen, RPMI 1640 medium, and PBS were obtained from Gibco; penicillin-streptomycin double antibiotic were obtained from Solarbio (Beijing, China); CL and their control liposome (PBSL) were obtained from LIPOSOMA (Netherlands); blocking agents CD16/CD32 Monoclonal Antibody (93) (Cat. #14-0161-82, 1:100), F4/80 Monoclonal Antibody (BM8), PE-eFluor 610 (Cat. #61-4801-82, 1:20), CD64 Monoclonal Antibody (X54-5/7.1), PerCP-eFluor 710 (Cat. #46-0641-82, 1:50), MERTK Monoclonal Antibody (DS5MMER), Alexa Fluor 700 (Cat. #56-5751-82, 1:20), Dendritic Cell Marker DCIR2 Monoclonal Antibody (33D1), PerCP-eFluor 710 (Cat. #46-5884-80, 1:100), CD4 Monoclonal Antibody (GK1.5), PerCP-eFluor 710 (Cat. #46-0041-80, 1:160), Ly-6C Monoclonal Antibody (HK1.4), PE-Cyanine7 (Cat. #25-5932-80, 1:160), CD8a Monoclonal Antibody (53–6.7), Alexa Fluor™ 700 (Cat. #56-0081-80, 1:160) were obtained from eBioscience; and APC anti-mouse/human CD11b Antibody (Cat. #101211, 1:100), Biotin anti-mouse CD192 (CCR2) Antibody (Cat. #150623, 1:100), Pacific Blue™ anti-mouse CX3CR1 Antibody (Cat. #149037, 1:200,), FITC anti-mouse CD169 (Siglec-1) Antibody (Cat. #142406, 1:100), PE anti-mouse CD11c Antibody (Cat. #117307, 1:100), Pacific Blue™ anti-mouse CD24 Antibody (Cat. #101820, 1:50), APC/Cyanine7 Streptavidin (Cat. #405208, 1:300), Alexa Fluor® 700 anti-mouse Ly-6G Antibody (Cat. #127621, 1:200), APC/Cyanine7 anti-mouse CD8a Antibody (Cat. #100713, 1:20), APC/Cyanine7 anti-mouse I-A/I-E Antibody (Cat. #107627, 1:100), Pacific Blue™ anti-mouse CD4 Antibody (Cat. #100427, 1:50), Biotin anti-mouse DC Marker (33D1) Antibody (Cat. #124903, 1:200), APC/Cyanine7 anti-mouse Ly-6C Antibody(Cat. #128025, 1:100), PE/Cyanine7 anti-mouse CX3CR1 Antibody (Cat. #149015, 1:1000) were obtained from BioLegend; RPMI 1640 medium containing 10% FBS, MACS buffer, etc., was prepared in-house.

## 2.3 Preparation of single-cell suspensions from mouse spleens

Spleens from the mice were placed in precooled RPMI 1640 medium supplemented with 10% FBS, minced, and digested with collagenase type VI (1 mg/mL) at 37°C in a 5% CO2, saturated humidity incubator for 30 minutes. The digestion was stopped by adding excess complete medium (RPMI 1640 with 10% FBS) containing 2 mM EDTA to chelate calcium and magnesium ions, thereby preventing cell aggregation. We filtered the digestion products through a 40 µm filter and centrifuged at 1800 r/min for 5 minutes at 4°C, after which the red blood cells were lysed with ACK buffer at room temperature for 2 minutes. After centrifugation, the cells were resuspended in 10% FBS in RPMI 1640 medium and kept on ice until use.

## 2.4 Labeling of cell surface molecular fluorescent antibodies

The cells were resuspended in MACS buffer to a concentration of $5 \times 10^7$/mL. One hundred microlitres of the suspension was transferred, and 1 µL of CD16/CD32 blocking reagent was added. The mixture was incubated on ice in the dark for 5 min. Fluorochrome- or biotin-conjugated antibodies (dose adjusted on the basis of fluorescence intensity) were added, mixed well, and incubated on ice in the dark for an additional 15 min. The cells were washed twice with MACS buffer by centrifugation (1800 rpm, 5 min each), and the supernatants were discarded.

 Direct labeling: The cells were resuspended in MACS buffer and analyzed directly via flow cytometry.

 Indirect labeling: For two additional washes, fluorochrome-conjugated streptavidin (SA) was added, the samples were incubated on ice in the dark for 15 min, washed twice again, resuspended in MACS buffer, and analyzed via a multicolor flow cytometer.

## 2.5 Injection of PBSL and CL

Each experimental group included 3 mice (three independent experiments were performed, eachwith 3 mice per group (total n = 9 mice)). The mice were intraperitoneally injected with 200μLof PBSL or CL (at a concentration of 5 mg/mL,resulting in a dose of 1 mg per mouse), and the spleens were harvested for analysis 3 days later.

## 2.6 Flow cytometry data analysis

Flow cytometry data were analyzed via FlowJo 10.10 software. Viable leukocytes were identified and gated based on forward scatter (FSC-A) and side scatter (SSC-A) properties to exclude cell debris and dead cells, which typically exhibit low FSC and aberrantly high SSC signals. This morphological gating strategy is an established method for enriching live cell populations in flow cytometry [19]. Prior to analysis, fluorescence compensation was performed using single-stained controls for each fluorochrome to correct for spectral overlap, as per standard multicolor flow cytometry practice. All flow cytometry experiments were based on 3 independent repetitions (each containing 3 mice).

## 2.7 Statistical analysis

Statistical analyses were performed with GraphPad Prism 8.01 (GraphPad Software, La Jolla, CA, USA). Normality and variance homogeneity were assessed by Shapiro-Wilk and Bartlett's tests, respectively. Welch's ANOVA + Games-Howell post-hoc test was used for normal data with unequal variances; Kruskal-Wallis H test + Dunn's post-hoc test for non-normal data. Descriptive statistics were used for experiments without intergroup comparisons. Significance was set at $P < 0.05$ ($P < 0.05$: *, $P < 0.01$: **, $P < 0.001$: ***, $P < 0.0001$: ****, $P \geq 0.05$: ns).

## 3 Results

### 3.1 Identification and characterization of TZM-like macrophage subsets within the splenic cDC2 gate

Leveraging the structural homology between splenic white pulp and lymph node TCZ, we investigated whether lymph node-resident scavenger macrophage (TZM) counterparts exist within the splenic cDC2 compartment using multi-parametric flow cytometry. Employing classical CD11c$^{high}$MHCII$^{high}$ gating integrated with functional markers, we identified a CD169$^-$CD24$^+$CD64$^+$CX3CR1$^+$MERTK$^+$DCIR2$^+$subset within F4/80$^{inter-low}$ cDC2s (Fig 1A), representing 0.59% of total splenic cDC2s. This population exhibited minimal CCR2 and Ly6C expression (Fig 1A), indicating a non-monocytic ontogeny. CL depletion triggered a > 10-fold increase in its relative frequency versus PBSL controls ($p < 0.0001$; Figs 1B-1D)—a consequence of selective depletion of CL-sensitive cells, which elevated the proportion of resistant subsets. This confirms depletion resistance characteristic of lymphoid tissue-resident macrophages [16,20]. Coexpression of CX3CR1 and MERTK—core mediators of apoptotic clearance in TZMs [16]—suggests a functional potential consistent with scavenger macrophages; however, direct experimental evidence for apoptotic cell clearance by this subset awaits future validation through specific phagocytosis assays. The combined evidence of CL resistance and absence of migration markers (e.g., CCR7) defines a novel putative tissue-resident entity embedded within splenic cDC2s. This phenotypic and depletion-based characterization challenges conventional classification frameworks and necessitates refined gating strategies to prevent misclassification of myeloid subsets.

### 3.2 Identification and phenotypic characterization of a resident F4/80$^{high}$ APC-like subset

To identify non-migratory counterparts of migratory F4/80$^{high}$ APCs in splenic cDC2s, we assessed CL depletion sensitivity and monocyte marker expression. This approach revealed an F4/80$^{high}$CD169$^+$subset (Fig 2A) that exhibits phenotypic similarity to migratory APCs but displays definitive residency characteristics. Specifically, its CCR2$^-$Ly6C$^-$ subpopulation—representing 13.3% of F4/80$^{high}$ cells and 2.7% of total cDC2—demonstrated complete CL-sensitivity ($p < 0.0001$ versus PBSL controls; Figs 2B-2C), consistent with classical APCs behavior [15,21] and contrasting tissue-resident

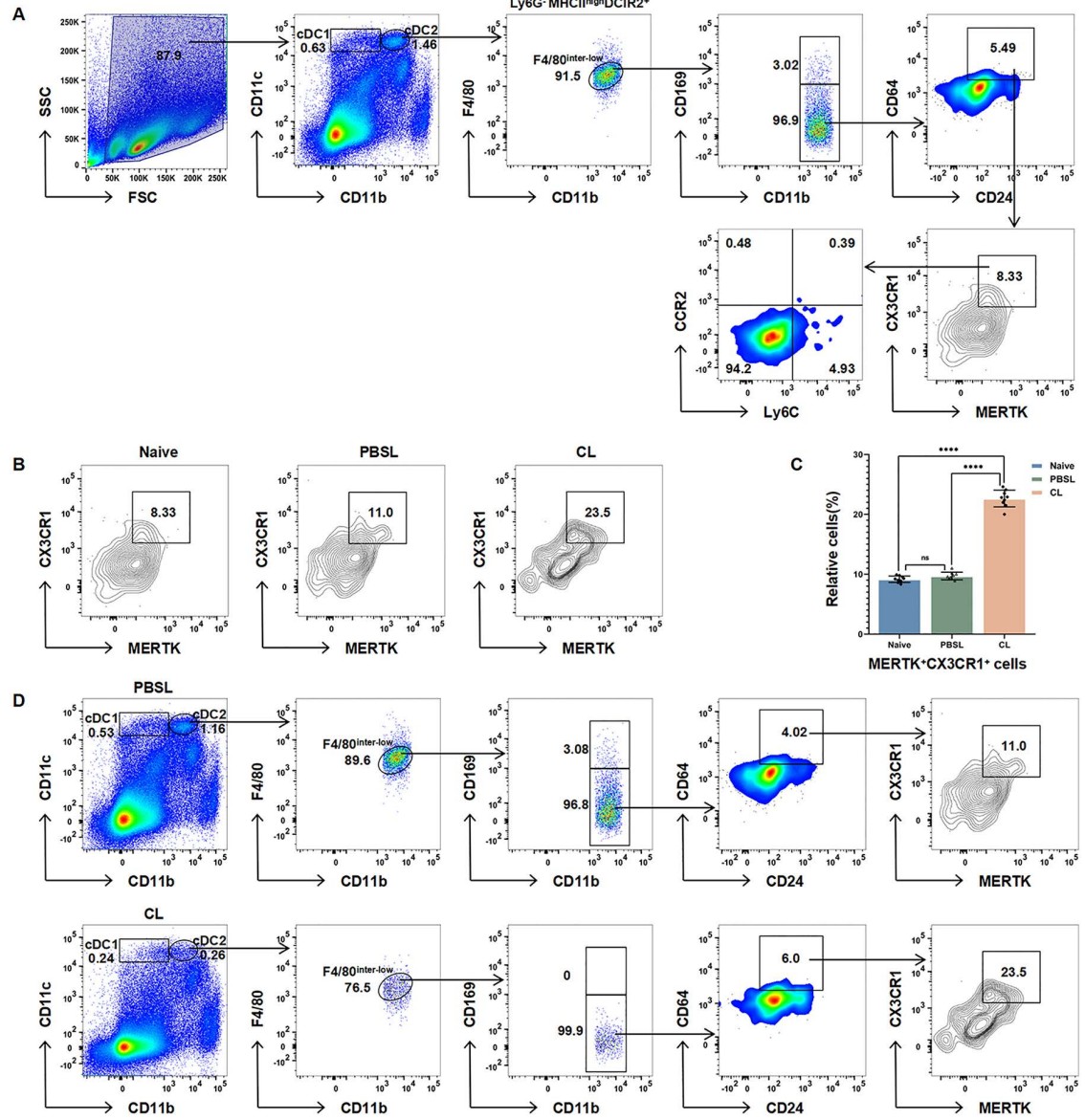

**Fig 1. Identification and functional analysis of novel macrophage-like subsets within the cDC2 gate of the mouse spleen. (A)** Phenotypic analysis by multicolor flow cytometry revealed the presence of an F4/80$^{inter-low}$CD169$^-$CD24$^+$CD64$^+$CX3CR1$^+$MERTK$^+$DCIR2$^+$ cell subset within the cDC2 gate of naive mouse spleens, which shares highly similar phenotypic features with lymph node TZMs. **(B)** CL depletion experiments revealed a significant increase in the proportion of TZM-like cells in the CL-treated group compared with the control group (PBSL-treated), suggesting macrophage-specific resistance to depletion (n = 9, repeated independently 3 times). **(C)** Quantification of the relative frequency (proportion) of TZM-like cells within the cDC2 gate. **(D)** Dynamic changes in the proportions of subsets within the cDC2 gate after PBSL and CL intervention. Data are presented as mean ± SD (n = 9). Normality test confirmed normal distribution, and Bartlett's test indicated heterogeneous variance; analyzed by Welch's one-way ANOVA with Games-Howell post-hoc test. ****$P < 0.0001$; ns (not significant).

macrophages. The absence of monocyte-derived markers (CCR2/Ly6C) suggests a unique differentiation pathway potentially specialized for tissue-localized immune functions. These findings define a novel phenotypically distinct APC-like subset within the splenic cDC2 compartment, expanding current paradigms of APC heterogeneity. Its definitive antigen-presenting capacity remains to be validated through functional assays such as T cell activation.

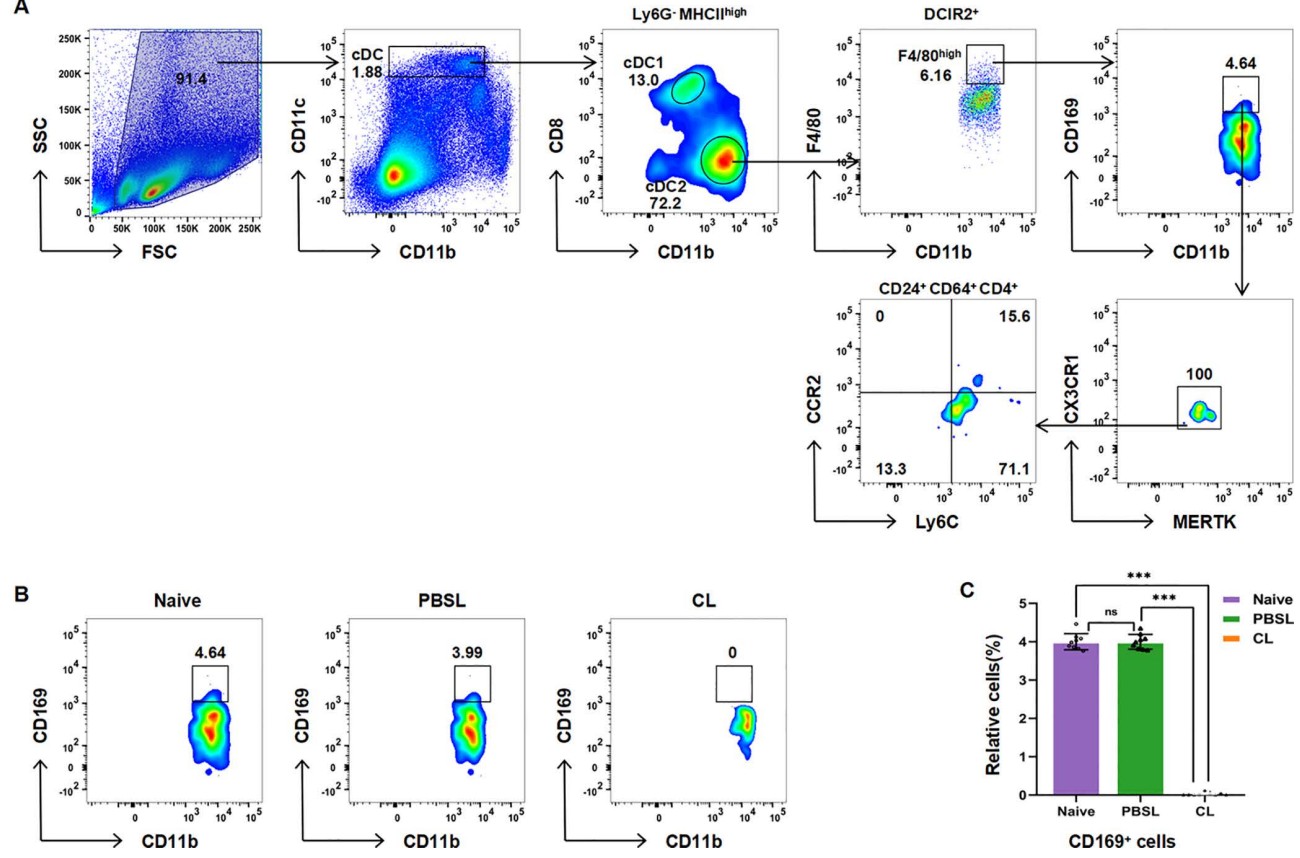

**Fig 2. Identification of resident F4/80$^{high}$ APC-like subsets within the cDC2 gate of the mouse spleen. (A)** Flow cytometric phenotypic analysis revealed the presence of an F4/80$^{high}$CD169$^+$CD24$^+$CD64$^+$CD4$^+$CX3CR1$^+$MERTK$^+$DCIR2$^+$ cell subset within the cDC2 gate, revealing heterogeneity in CCR2/Ly6C expression. **(B)** The proportion of F4/80$^{high}$ APC-like cells was significantly reduced in the CL-treated group, with no change in the PBSL control group (n = 9, repeated independently 3 times). **(C)** Quantitative statistics of the proportion of CD169$^+$ cells. Data are presented as median (inter-quartile range) (n = 9). Normality test indicated non-normal distribution; analyzed by Kruskal-Wallis H test with Dunn's post-hoc test. ***$P < 0.001$; ns (not significant).

### 3.3 Identification and functional analysis of CD4/CD8α biphenotypic cells within splenic cDC subsets

Traditional models classify splenic cDCs as CD8α$^+$cDC1s or CD8α$^-$cDC2s (with cDC2s subdivided into CD4$^+$/CD4$^-$ sub-sets) [12,22]. However, our high-dimensional flow cytometry analysis identified CD4$^+$CD8α$^+$ dual-positive cDCs that challenge this paradigm. Within this biphenotypic population, cDC1s represented 12.1% and cDC2s constituted 1.9% (Figs 3A-3C), alongside nonclassical subsets such as CD4$^+$CD8α$^-$cDC1s (0.6%) (Figs 3B-3C) and CD4$^-$CD8α$^+$cDC2s accounted for 1.03% of splenic cDC2s (Fig 3D). Further analysis revealed CD4$^+$CD8α$^+$ (2.57%) subsets within the cDC2 gate (Fig 3D). Notably, F4/80$^{high}$ APC-like cells showed predominant enrichment in CD4$^+$CD8α$^+$cDC2s (Figs 3A-3B,3D), with minor presence in CD4$^+$CD8α$^-$ subsets (Fig 3B). Since CD4$^+$cDC2 localize to niches associated with Th2/Th17 responses [2,7], this distribution suggests a potential role in T helper cell polarization; however, this functional bias remains speculative and requires direct validation through cytokine profiling and T cell polarization assays. Collectively, these findings provide new insights into cDC-macrophage functional convergence. While these phenotypic profiles suggest functional plasticity, direct evidence of MHC presentation capabilities in biphenotypic subsets awaits future validation through functional assays such as OVA cross-presentation.

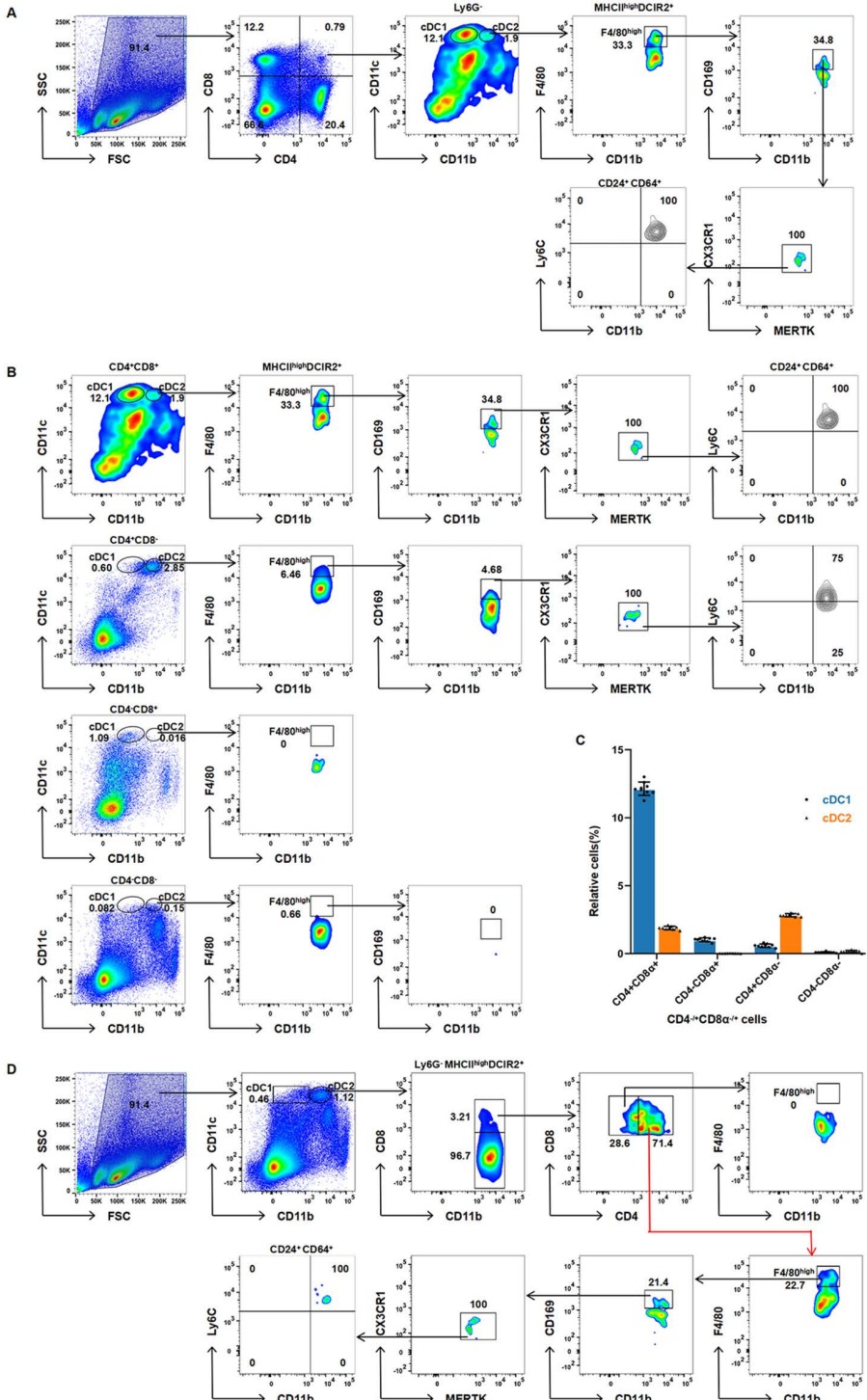

**Fig 3. Identification and distribution of CD4/CD8α⁺ biphenotypic subsets within the cDC gate of the mouse spleen. (A)** Flow cytometry revealed the coexistence of the CD4⁺CD8α⁺ cDC1 and cDC2 subsets. **(B)** Distribution proportions of cDC1s and cDC2s in different CD4/CD8α expression combinations, with the gray box highlighting the specific enrichment of F4/80^high APC-like cells. **(C)** Statistical analysis of the proportions of CD4^+/-CD8α^+/- cDC1s and cDC2s. **(D)** Quantitative analysis of CD4⁻CD8α⁺ (1.03%) and CD4⁺CD8α⁺ (2.57%) subsets within the cDC2 gate. Data are presented as descriptive statistics (n = 9), showing the distribution characteristics of specific cell subsets without intergroup statistical comparison.

## 4 Discussion

This study systematically deciphers the phenotypic complexity of splenic CD11c$^{high}$MHCII$^{high}$ cDCs through integrated flow cytometry and clodronate depletion approaches. Our findings collectively redefine splenic cDC heterogeneity (Fig 4), revealing three novel subsets that revise conventional paradigms and suggest microenvironment-driven plasticity. We establish three fundamental revisions to current paradigms: First, the identification of TZM-like macrophages (CX3CR1$^+$MERTK$^+$) within the cDC2 compartment, exhibiting profound CL resistance (>10-fold frequency increase versus controls, $p<0.0001$; Fig 1) and functional parallels to lymphoid-resident scavengers [16]; Second, the discovery of a novel tissue-resident APC lineage (F4/80$^{high}$CCR2$^-$Ly6C$^-$) distinguished by clodronate sensitivity (Fig 2) and absence of monocyte markers, challenging conventional migratory APC classifications; Third, the documentation of CD4$^+$CD8α$^+$ biphenotypic hybrids coexpressing markers across 12.1% of cDC1s and 1.9% of cDC2s (Fig 3), directly contradicting mutually exclusive CD4/CD8α dichotomies [12].

This study delivers a fundamentally revised discriminative framework for splenic cDC classification by integrating three novel subsets: (1) TZM-like macrophages (CX3CR1$^+$MERTK$^+$, clodronate-resistant), (2) resident F4/80$^{high}$ APCs (CCR2$^-$Ly6C$^-$, clodronate-sensitive), and (3) CD4$^+$CD8α$^+$ biphenotypic hybrids. These subsets redefine cDC heterogeneity beyond classical transcriptional paradigms and establish phenotype-driven taxonomy as a critical tool for resolving myeloid ambiguity. The core innovation lies in providing the first experimental evidence for TZM-like macrophages within the splenic cDC2 gate—a population exhibiting >10-fold clodronate resistance ($p<0.0001$; Fig 1)—and a resident APC lineage distinguished by CCR2$^-$Ly6C$^-$ expression and clodronate sensitivity (Fig 2). While functional

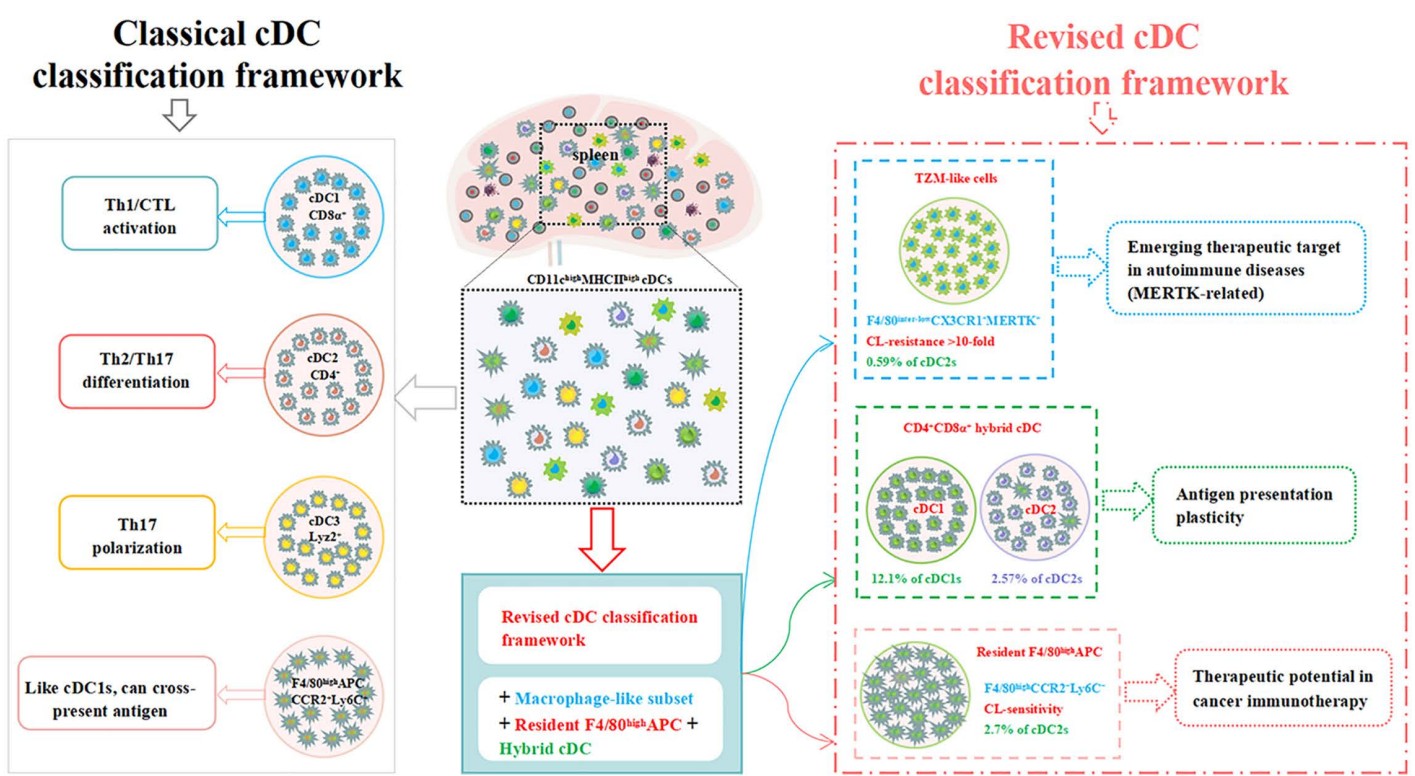

**Fig 4. Revised model of murine splenic cDC heterogeneity integrating novel subsets.** Schematic proposes three key revisions: (1) TZM-like macrophages (CX3CR1$^+$MERTK$^+$, clodronate-resistant) within cDC2 gate; (2) Resident F4/80$^{high}$ APCs (CCR2$^-$Ly6C$^-$, clodronate-sensitive); (3) CD4$^+$CD8α$^+$ biphenotypic hybrids in cDC1/cDC2 compartments. Arrows indicate phenotypic plasticity across subsets.

validation of biphenotypic cells (e.g., MHC presentation) remains for future studies, their unambiguous identification (12.1% cDC1s and 1.9% cDC2s; Fig 3) challenges the CD4/CD8α dichotomy and establishes a quantitative baseline for plasticity research. Collectively, these findings demonstrate microenvironment-driven plasticity of cDCs. Although the MHC presentation functionality of biphenotypic cells awaits validation (e.g., through cross-presentation assays in OVA models), their phenotypic profile suggests functional plasticity likely originates from local signals such as IFN-γ/TGF-β [6]. This plasticity may confer dual MHC presentation capabilities—a trait observed in hybrid APCs [15]—though functional validation in this specific context remains necessary. Our study deliberately focused on resolving phenotypic ambiguities through surface marker profiling and depletion assays. Although spatial localization and functional plasticity of the identified subsets represent essential future directions, the current work establishes a revised taxonomic framework centered on CX3CR1/MERTK coexpression and clodronate sensitivity. Therapeutically, MERTK$^+$ TZM-like macrophages provide a mechanistic link to systemic lupus erythematosus [23], while CCR2$^-$ Ly6C$^-$ resident APCs may serve as endogenous compensators in APC-deficient tumors [24]. Though functional validation in disease models is needed, our phenotypic map delivers essential 'coordinates' for targeting these subsets. Conversely, the resident APC lineage displays paradoxical CL-sensitivity akin to migratory APCs [15] yet demonstrates tissue residency, suggesting compensatory potential in APC-deficient tumors like melanoma [24].

Critically, our surface-marker-centric approach (CX3CR1/MERTK coexpression + clodronate sensitivity) offers an actionable tool to prevent myeloid misclassification. Traditional CD11c$^{high}$MHCII$^{high}$ gating overlooks transitional subsets, as demonstrated by the misplacement of TZM-like macrophages into cDC2s. This revised framework enables precise dissection of cDC plasticity in future studies, from developmental trajectories to disease-associated reprogramming. Our work further exposes critical limitations of classical CD11c$^{high}$MHCII$^{high}$ gating strategies, which overlook precursor populations (e.g., Ly6C$^+$DC progenitors [2]) and misclassify phenotypically ambiguous subsets [16]. Future investigations should prioritize spatial transcriptomics to resolve three key dimensions: (1) developmental trajectories of biphenotypic cDCs, (2) apoptotic clearance mechanisms by macrophage-like subsets, and (3) synergistic cDC2/cDC3 interactions in tumor microenvironments.

While the sample size (n = 9 mice) provided statistical rigor for initial discovery under steady-state conditions, validation across disease models necessitates expanded cohorts. Similarly, functional confirmation—such as cross-presentation assays in OVA models—remains essential. Nevertheless, this study delivers a refined discriminative framework centered on CX3CR1/MERTK coexpression and clodronate sensitivity/resistance (Figs 1–2), effectively preventing myeloid subset misclassification.

While our study provides a detailed phenotypic landscape and resolves subset ambiguity, it has certain limitations. First, the definitive APC functionality of the identified subsets remains to be experimentally validated. Second, the analysis was conducted in young adult mice (7–8 weeks old), and age-related dynamics are unexplored. Third, our approach did not include spatial transcriptomics or immunohistochemistry; thus, the precise anatomical localization of these novel subsets within the splenic architecture remains a critical open question. Fourth, the designation of certain subsets as "resident" or having "migratory potential" is inferred from marker profiles (e.g., absence of CCR7) and differential sensitivity to clodronate liposome depletion. Definitive proof through direct methods, such as cell labeling and tracking in vivo, will be an important focus of future work. Fifth, this phenotypic snapshot does not directly address the ontogenic relationships or developmental trajectories among these novel subsets and classical cDCs. Fate-mapping and lineage-tracing studies will be valuable to determine whether they arise from common or distinct progenitors. Sixth, this study has technical limitations in flow cytometry analysis. Specifically, viable cell gating relied on morphological FSC-A/SSC-A parameters without fluorescent viability staining, and the critical FSC-H or FSC-W parameters necessary for standard singlet gating were not recorded. Although stringent morphological gating and optimized sample handling were applied to mitigate these concerns, future studies will incorporate both viability dyes and FSC-A/FSC-H (or FSC-W) singlet gating as essential parameters.

Future studies integrating functional assays (including T cell activation and polarization assays to define Th-cell bias), aging models, spatial techniques, lineage-tracing approaches, detailed ontogenic analysis and interrogation of subset stability and behavior under inflammatory conditions will be essential to fully understand the biology of these cells. Nevertheless, the phenotypic criteria established here provide a robust phenotypic framework and essential 'coordinates' for future investigations into their development, functional specialization, and therapeutic targeting in immunity and disease.

In summary, we redefine splenic cDC heterogeneity through the integrative lens of TZM-like macrophages, a novel resident APC lineage, and plasticity-driven biphenotypic hybrids. This taxonomy not only revises foundational classification systems but also opens targeted therapeutic avenues in cancer immunotherapy and autoimmune disease modulation.

## Supporting information

**S1 File. Mouse Flowcytometry cellsubsets three groups.**
(XLSX)

## Acknowledgments

We thank Ruihan Guo from Hanyang University for his assistance with statistical analysis and figure optimization.

## Author contributions

**Conceptualization:** chunqing Yang, Qinqin Wang.

**Data curation:** Qingjie Xue, Yu Feng, Wenjun Ding, Ying Lu.

**Formal analysis:** chunqing Yang, Qinqin Wang.

**Funding acquisition:** Qinqin Wang.

**Investigation:** chunqing Yang, Yu Feng.

**Methodology:** chunqing Yang, Yu Feng, Qinqin Wang.

**Project administration:** chunqing Yang, Qinqin Wang.

**Software:** Yu Feng, Wenjun Ding, Ying Lu.

**Supervision:** chunqing Yang.

**Validation:** chunqing Yang.

**Writing – original draft:** chunqing Yang, Qingjie Xue, Qinqin Wang.

**Writing – review & editing:** chunqing Yang, Qinqin Wang.

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
