## [Decision Letter · Decision Letter 0]

22 Oct 2025

Dear Dr. Yang,

Thank you for submitting your manuscript to PLOS ONE. After careful consideration, we feel that it has merit but does not fully meet PLOS ONE’s publication criteria as it currently stands. Therefore, we invite you to submit a revised version of the manuscript that addresses the points raised during the review process.

Thank you for submitting your manuscript titled " High-Dimensional Phenotyping Reveals Novel Macrophage-Like and Hybrid Subsets within Murine Splenic Conventional Dendritic Cells" to *PlosOne* . Although the subject matter of your paper is of interest, a number of concerns were raised by the both reviewers. While these concerns preclude publication of this manuscript in its current form, you are invited to resubmit an appropriately revised manuscript that addresses the reviewers' concerns.

Please see the comments below, which contain a summary of the concerns raised by the reviewers. Please provide a point-by-point reply that indicates how you have revised your manuscript to address each of these concerns.

We look forward to receiving your revised manuscript.

Kind regards,

Subhasis Barik

Academic Editor

PLOS ONE

**Journal Requirements:**

1. When submitting your revision, we need you to address these additional requirements. Please ensure that your manuscript meets PLOS ONE's style requirements, including those for file naming. The PLOS ONE style templates can be found at https://journals.plos.org/plosone/s/file?id=wjVg/PLOSOne_formatting_sample_main_body.pdf and https://journals.plos.org/plosone/s/file?id=ba62/PLOSOne_formatting_sample_title_authors_affiliations.pdf 2. We note that the grant information you provided in the ‘Funding Information’ and ‘Financial Disclosure’ sections do not match.  When you resubmit, please ensure that you provide the correct grant numbers for the awards you received for your study in the ‘Funding Information’ section. 3. Thank you for stating in your Funding Statement: This work was supported by the Natural Science Foundation of Shandong Province (ZR2024MH129), the National Nature Science Foundation of China (82301629) and Project of the Innovative Training Program for Students at Jining Medical University (cx2020118).   Please provide an amended statement that declares *all* the funding or sources of support (whether external or internal to your organization) received during this study, as detailed online in our guide for authors at http://journals.plos.org/plosone/s/submit-now.  Please also include the statement “There was no additional external funding received for this study.” in your updated Funding Statement. Please include your amended Funding Statement within your cover letter. We will change the online submission form on your behalf. 4. We note that your Data Availability Statement is currently as follows: All relevant data are within the manuscript and its Supporting Information files. Please confirm at this time whether or not your submission contains all raw data required to replicate the results of your study. Authors must share the “minimal data set” for their submission. PLOS defines the minimal data set to consist of the data required to replicate all study findings reported in the article, as well as related metadata and methods (https://journals.plos.org/plosone/s/data-availability#loc-minimal-data-set-definition). For example, authors should submit the following data: - The values behind the means, standard deviations and other measures reported;- The values used to build graphs;- The points extracted from images for analysis. Authors do not need to submit their entire data set if only a portion of the data was used in the reported study. If your submission does not contain these data, please either upload them as Supporting Information files or deposit them to a stable, public repository and provide us with the relevant URLs, DOIs, or accession numbers. For a list of recommended repositories, please see https://journals.plos.org/plosone/s/recommended-repositories. If there are ethical or legal restrictions on sharing a de-identified data set, please explain them in detail (e.g., data contain potentially sensitive information, data are owned by a third-party organization, etc.) and who has imposed them (e.g., an ethics committee). Please also provide contact information for a data access committee, ethics committee, or other institutional body to which data requests may be sent. If data are owned by a third party, please indicate how others may request data access. 5. Please upload a new copy of Figures 1, 2 and 3, as the detail is not clear. Please follow the link for more information:  https://journals.plos.org/plosone/s/figures 6. If the reviewer comments include a recommendation to cite specific previously published works, please review and evaluate these publications to determine whether they are relevant and should be cited. There is no requirement to cite these works unless the editor has indicated otherwise. 

**Additional Editor Comments:**

Although the subject matter of your paper is of interest, a number of concerns were raised by the both reviewers. While these concerns preclude publication of this manuscript in its current form, you are invited to resubmit an appropriately revised manuscript that addresses the reviewers' concerns.

Reviewers' comments:

**Comments to the Author**

1. Is the manuscript technically sound, and do the data support the conclusions?

Reviewer #1: Partly

Reviewer #2: Partly

2. Has the statistical analysis been performed appropriately and rigorously?

Reviewer #1: No

Reviewer #2: No

3. Have the authors made all data underlying the findings in their manuscript fully available?

Reviewer #1: Yes

Reviewer #2: No

4. Is the manuscript presented in an intelligible fashion and written in standard English?

Reviewer #1: Yes

Reviewer #2: Yes

**Reviewer #1:** The manuscript entitled “High-Dimensional Phenotyping Reveals Novel Macrophage-Like and Hybrid Subsets within Murine Splenic Conventional Dendritic Cells” uses multiparametric flow cytometry to discover hitherto uncharacterized subsets of conventional DCs within the mouse spleen. The work, as a whole, is a very primitive one and does not have much to offer towards the advancement of knowledge in its context. I would specifically like the authors to work on the following areas to improve the quality of the manuscript.

1- The work only explores the immunophenotypes of the identified cDC subsets, but does not look into their functional relevance. How, then, do the authors confirm that the cDCs are indeed acting as APC? This question becomes even more relevant as the authors claim to have "functionally verified" the cDC subsets, but have only tested their sensitivity to liposomal clodronate. Liposomal clodronate does not directly guarantee their antigen presenting capacity, and is only an indicator of their phagocytic activity. Additional experiments demonstrating the antigen presentation capacity of these cDCs (like surface MHC expression, ability to trigger TCR-proximal signaling in T cells etc) must also be done if the authors prefer sticking to the statement that they have “functionally verified” the cDC subsets.

2- Furthermore, all experiments are conducted only in 7-8 weeks old mice, and do not explore the age-related variations in the frequencies and distributions of these cDC subsets. Given the plasticity of the splenic microenvironment with age, what are the chances that the patterns may be age-specific?

3- Do these different cDC subsets have different spatial distributions within the spleen?

4- The assumptions that certain cDC subsets are resident or migratory are completely based on their immunophenotypes, without sufficient definitive proof. Authors should adopt requisite validatory approaches by using cell-labeling and tracking.

5- Do these different cDC subsets have any ontogenic relation among them? Authors might consider rechecking the flow cytometry data from their clodronate experiment to check whether the depletion of any one of the subsets leads to a compensatory increase in any other subsets; or causes an increase in the expression of one or more of the immunophenotypic markers corresponding to the depleted subsets within the surviving subsets. This might give important insights into possible ontogenic relations among them.

6- The exact amount of liposomes injected (in terms of mass or molarity, not volume) needs to be reported.

7- Statistical tests done against each experiment need to be mentioned in the respective figure legends. Moreover, the specific post-hoc tests for ANOVA need to be mentioned.

8- Most of the gating schemes show an absence of a singlet gating via FSC-A vs FSC-H combinations. How do the authors ensure that no doublets are present in their samples during flow cytometric acquisition?

9- On the graphical data corresponding to experimental replicates, individual values need to be plotted alongside the measures of central tendency and dispersion, to show the exact distribution of data points.

**Reviewer #2:** The research article entitled “High-Dimensional Phenotyping Reveals Novel Macrophage-Like and Hybrid Subsets within Murine Splenic Conventional Dendritic Cells” attempts to uncover and reclassify a unique subset of murine splenic conventional dendritic cells using multi-parametric flow cytometry alone. As a whole, the work has the potential to create a future breakthrough, but on the other hand, it is very superficial at the present time.

I would like to shed some light in this regard, drawing the author's attention to some immediate points to reconsider.

1. The identified CD169⁻CD24⁺CD64⁺CX3CR1⁺MERTK⁺DCIR2⁺ subset within F4/80inter-low cDC2s is correlated with the co-expression of CX3CR1 and MERTK on TZM and claimed for their functional role in apoptotic clearance, just with the cell surface-based markers, without any functional experimental proof under in vitro or ex vivo settings.

In vitro experiment, such as coculturing the designated sorted cells with fluorescent dye (eg, Hoechst, pHrodo, Annexin A5-pHrodo) labelled apoptotic cells, followed by flow cytometric detection of the labelled dye within the gate of designated cells, will prove their true role in apoptotic clearance. Similarly designated cells could be isolated from mice injected with labelled apoptotic cells and detected for the fluorescent signal for ex vivo validation.

2. The identified subset within F4/80inter-low cDC2s has been considered tissue resident, with the absence of cell migration-associated surface markers like CCR7; however, no experimental data can be observed in the manuscript.

3. Again, being a splenic residential cell (claimed by the authors), it is quite likely to get exposed to various inflammatory stimuli or inflammatory mediators, which may induce migration-associated surface markers involved in migration, if not, at least within the intra-splenic regions. Experimental validation of the same under inflammatory conditions will uncover its true tissue-resident status.

4. The authors claimed the F4/80ʰⁱᵍʰCD169⁺ subset as a novel resident APC, only with phenotypic characterisation without any valid functional experiments in this regard. Coculture-based experiments could be performed with designated cell population with murine splenic T-cells, isolated from mice injected with any antigen of choice (eg, OVA) to validate their efficiency as APC. Based on the readouts like MHC-II expression status, T-cell activation and cytokine profiling, designated resident cells will qualify to meet the standards of APC.

5. F4/80ʰⁱᵍʰCD4⁺CD8α⁺ cDC2s role in Th-biasness should be validated experimentally.

These are some key points to be considered, especially when the authors use words like “functional verification” and “characterisation”, along with identification.

I would also like to highlight a few more technical points for the authors.

1. The flow-cytometric gating strategy described in the manuscript figure didn’t ensure that the cells are in a living (7-AAD vs SSC plot missing) and singlet state (FSC-A vs FSC-H plot missing).

2. All animals considered for the experiments were from a similar age group, that is 7–8 weeks old C57BL/6 mice. This prevents deciphering the percentage or fraction of designated macrophage-like cDC throughout the growth of mice and susceptibility or biasness towards a particular inflammatory outcome, along with increasing age.

**Do you want your identity to be public for this peer review?** For information about this choice, including consent withdrawal, please see our Privacy Policy

Reviewer #1: No

Reviewer #2: No

---

## [Author Response · Author response to Decision Letter 1]

24 Dec 2025

To the Academic Editor and Reviewers,

Thank you for your constructive feedback and for granting an extension to complete the revisions. We have thoroughly addressed all points raised, significantly strengthening the manuscript. Below is a summary of the key revisions made:

Journal Requirements:

1. Formatted the manuscript to meet PLOS ONE’s style requirements, including file naming and document structure.

2. Updated the funding statement and reconciled it with the Financial Disclosure section.

3. Added a Data Availability Statement; the complete analytical dataset is provided as Mouse_FlowCytometry_CellSubsets_ThreeGroups and will be uploaded as Supporting Information.

4. Re-exported all figures (Figures 1–3) as high-resolution TIFF files compliant with PLOS ONE guidelines.

5. Reviewed and reformatted the reference list, removed invalid citations, and added the recommended flow cytometry guidelines citation.

Reviewer Comments:

We have carefully moderated our claims and expanded the discussion of limitations in direct response to the reviewers’ critiques:

1. Functional validation: We have toned down claims of “functional verification” and clarified that the current study is primarily phenotypic. Future assays (e.g., cross-presentation, T-cell activation) are explicitly highlighted in the Discussion and Limitations sections.

2. Study scope and limitations: We have added dedicated statements acknowledging key limitations, including the need to investigate age-related dynamics, spatial localization, definitive proof of residency/migration, and ontogenic relationships.

3. Methodological rigor: The CL dose (1 mg per mouse) is now specified in Methods. A critical limitation regarding the absence of standard singlet gating (FSC-H/FSC-W parameters not recorded) is transparently disclosed, along with compensatory measures applied.

4. Statistical and data presentation: All figure legends now specify statistical tests used, and individual data points are included in Figures 1 and 2. The statistical approach is detailed in a new Section 2.7.

5. Phenotypic inferences: Claims regarding tissue residency, migratory potential, and functional roles (e.g., apoptotic clearance, Th-bias) have been moderated and framed as inferred from marker profiles, with direct experimental validation highlighted as future work.

We believe these revisions have enhanced the manuscript’s clarity, rigor, and balance, and we hope it now meets the high standards of PLOS ONE for publication.

Sincerely,

Chunqing Yang, Ph.D.

Corresponding Author

Institute of Mental Health, Jining Medical University

---

## [Decision Letter · Decision Letter 1]

13 Jan 2026

High-Dimensional Phenotyping Reveals Novel Macrophage-Like and Hybrid Subsets within Murine Splenic Conventional Dendritic Cells

PONE-D-25-48158R1

Dear Dr. Yang,

We’re pleased to inform you that your manuscript has been judged scientifically suitable for publication and will be formally accepted for publication once it meets all outstanding technical requirements.

Kind regards,

Subhasis Barik

Academic Editor

PLOS One

Additional Editor Comments (optional):

Authors have addressed all the concerns provided by the reviewers. Both the reviewers have aggreed that the manuscript is now suitable for publication.

Reviewers' comments:

Reviewer's Responses to Questions

**Comments to the Author**

Reviewer #1: All comments have been addressed

Reviewer #2: All comments have been addressed

2. Is the manuscript technically sound, and do the data support the conclusions?

Reviewer #1: Yes

Reviewer #2: Yes

3. Has the statistical analysis been performed appropriately and rigorously?

Reviewer #1: Yes

Reviewer #2: Yes

4. Have the authors made all data underlying the findings in their manuscript fully available?

Reviewer #1: Yes

Reviewer #2: Yes

5. Is the manuscript presented in an intelligible fashion and written in standard English?

Reviewer #1: Yes

Reviewer #2: Yes

Reviewer #1: Authors have met necessary corrections. The rectifications are satisfactory. I have no further issue with this manuscript.

Reviewer #2: Upon reviewing the revised manuscript, it is evident that the authors have addressed all the points raised with proper justification.

**Do you want your identity to be public for this peer review?** For information about this choice, including consent withdrawal, please see our Privacy Policy

Reviewer #1: No

Reviewer #2: No

---

## [Editor Report · Acceptance letter]

PONE-D-25-48158R1

PLOS One

Dear Dr. Yang,

I'm pleased to inform you that your manuscript has been deemed suitable for publication in PLOS One. Congratulations! Your manuscript is now being handed over to our production team.

Kind regards,

on behalf of

Dr. Subhasis Barik

Academic Editor

PLOS One